# Syntheses of Polypeptides and Their Biomedical Application for Anti-Tumor Drug Delivery

**DOI:** 10.3390/ijms23095042

**Published:** 2022-05-02

**Authors:** Huayang Feng, Jonas Fabrizi, Jingguo Li, Christian Mayer

**Affiliations:** 1Institute for Physical Chemistry, CeNIDE, University of Duisburg-Essen, 45141 Essen, Germany; huayang.feng@stud.uni-due.de (H.F.); jonas.fabrizi@stud.uni-due.de (J.F.); 2People’s Hospital of Zhengzhou University, Zhengzhou University, Zhengzhou 450003, China

**Keywords:** polypeptide synthesis, activated amino acid monomers, drug delivery, nanocarriers, cancer treatment

## Abstract

Polypeptides have attracted considerable attention in recent decades due to their inherent biodegradability and biocompatibility. This mini-review focuses on various ways to synthesize polypeptides, as well as on their biomedical applications as anti-tumor drug carriers over the past five years. Various approaches to preparing polypeptides are summarized, including solid phase peptide synthesis, recombinant DNA techniques, and the polymerization of activated amino acid monomers. More details on the polymerization of specifically activated amino acid monomers, such as amino acid N-carboxyanhydrides (NCAs), amino acid N-thiocarboxyanhydrides (NTAs), and N-phenoxycarbonyl amino acids (NPCs), are introduced. Some stimuli-responsive polypeptide-based drug delivery systems that can undergo different transitions, including stability, surface, and size transition, to realize a better anti-tumor effect, are elaborated upon. Finally, the challenges and opportunities in this field are briefly discussed.

## 1. Introduction

Cancer is the second leading cause of death worldwide. The efficacy of many anti-tumor drugs is often reduced via rapid blood clearance, non-specific biodistribution, or poor accumulation and retention in tumor sites [1]. In addition, many anti-tumor drugs have inherent limitations, such as poor water solubility and low cellular uptake [2]. Therefore, many kinds of drug delivery systems (DDS), based on polypeptides, polyesters, mesoporous silica, gold nanoparticles, etc., have been developed to address these deficiencies [3]. 

Among them, polypeptides have received extensive attention due to their innate biocompatibility and degradability [2]. When dispersed in water, polypeptides with hydrophilic/hydrophobic segments can form micelles, vesicles, hydrogels, and capsules [4,5]. Due to the above-mentioned versatile structures, and their biocompatibility and biodegradability, polypeptides are extensively studied as carriers for drug delivery [6]. With the development of a variety of well-controlled polymerization chemistries, polypeptides can be easily integrated into other materials to synthesize hybrid materials with even more versatile features for self-assembly and controlled release [2,4,7]. These polypeptides and polypeptide-based hybrid materials exhibit controlled drug-release properties because of their natural amino acid residues with innate stimuli-responsive characteristics or other responsive moieties from their hybrid materials [8].

The present review focuses on the synthesis of polypeptides through the polymerizations of NCA, NTA, and NPC monomers, as well as the recent anti-tumor drug delivery applications of polypeptide materials and their hybrids over the past five years. Finally, challenges and opportunities for the design and preparation of the polypeptide-based drug delivery systems for clinical translation are briefly discussed.

## 2. Synthesis of Polypeptides through Polymerizations of Activated Amino Acid Monomers 

Commonly, three different synthetic pathways are used to prepare peptides in the laboratory: via the polymerization of amino acid N-carboxyanhydrides (NCAs) [6,9,10], amino acid N-thiocarboxyanhydrides (NTAs) [11,12], and N-phenoxycarbonyl amino acids (NPCs) [13]; via various stepwise coupling reactions of α-amino acids, such as during solid phase peptide synthesis (SPPS) [14]; or via recombinant DNA techniques for expressing peptides in microorganisms [14]. Polypeptides that are synthesized through SPPS have controlled primary sequences and can fulfill certain functionalities, but it is difficult to create high molecular weight polypeptides above 100 residues, due to the inevitable side reactions [15]. Recombinant DNA techniques can create polypeptides with specific sequences and high molecular weights. Moreover, they allow for peptide production on a very large scale [16]. However, specialized equipment, which is not readily available in most synthetic laboratories, is needed for this method [6]. The polymerization of activated amino acid monomers enables the formation of bioactive and high molecular weight polypeptides in a facile and expedient manner. As shown in Figure 1, the process begins with the conversion of amino acids into the corresponding activated monomers; afterwards, polymerization is initiated in the presence of certain initiators. Although polypeptides that are synthesized in this way lack precise sequence control, their synthetic advantages make this method attractive and economical for synthesizing polypeptides in large quantities [6].

### 2.1. Ring-Opening Polymerization (ROP) of α-Amino Acid N-Carboxy Anhydrides (NCAs) 

During 1906 and 1908, Hermann Leuchs published three papers that described the synthesis and properties of α-amino acid N-carboxyanhydrides (NCAs, Figure 2a) [9]. NCAs were discovered by coincidence when Leuchs attempted the purification of N-ethoxycarbonyl α-amino acid chlorides. However, Leuchs changed his area of research completely from NCAs to the chemistry of strychnine alkaloids after 1907, due to the lack of proper analytical methods for NCA-polymerized products and the wrong estimation of their structure. As a result of his pioneering work, NCAs are commonly referred to as Leuchs’ anhydrides [9].

Presently, the most important and economical method for synthesizing NCAs is called the Fuchs–Farthing method, in which phosgene or its derivatives are used as a cyclizing agent (Figure 2b). In 1922, Friedrich Fuchs described the preparation of NCA of N-phenylglycine via the phosgenation of N-phenylglycine in an aqueous solution. Based on Fuchs’ reaction, A. C. Farthing made some modifications and prepared some other NCAs such as NCAs of glycine, DL-β-phenylalanine, L-leucine, etc., in 1950 [17]. Using this system, it is typical to prepare an NCA via the reaction of an α-amino acid with phosgene in ethyl acetate (also dichloromethane and dioxane) at elevated temperatures (~60 °C) under an inert atmosphere [18,19,20]. 

The ring-opening polymerization (ROP) of NCAs can produce ‘living’ polypeptides. The expression “living polymers” was first described by M. Szwarc in 1956, when he tried to synthesize polystyrene via the polymerization of styrene [21]. The term “living” mainly means that when the polymerization is terminated either by 100% conversion of the monomers, by cooling, or by precipitation, the reactive end group that is responsible for chain growth remains unchanged (alive). In this way, polymers with controlled lengths and low polydispersity can be prepared. Moreover, it is also possible to form block copolymers via the sequential addition of different monomers [22]. 

A suitable initiator is essential for the polymerization of NCAs. Because of the numerous reactive sites on the five-membered NCA ring, i.e., the 2- and 5-carbonyl groups, 3-NH and 4-CH, there is a wide range of initiators available for the initiation of NCA polymerizations, e.g., protonic nucleophiles, nonprotonic bases, metal salts, organometallics, transition metals, and their analogues of strong bases [12,22]. All of the initiators and reaction mechanisms are explained in detail in the reference [6]. Primary amines are presently one of the most common initiators for the ROP of NCAs, because of the following two reasons. Firstly, using primary amines as initiators could prepare polypeptides with a living end group (amino group). Secondly, the highly nucleophilic and sterically unhindered nature of primary amines allows them to initiate polymerization rapidly, resulting in smaller polydispersity values [9]. One possible mechanism of primary amines-initiated ROP of NCAs is shown in Figure 3. The polymerization involves three steps: carbonyl addition, ring opening, and decarboxylation. In a first step, a nucleophilic primary amine bearing a lone pair of electrons attacks the C_5_ carbon of the NCA to initiate ring opening. This is followed by a decarboxylation step and regeneration of the amine. Finally, the regenerated amine attacks the molecule of another NCA to increase the length of the polypeptide chain. This pathway allows for the synthesis of co-polypeptides with several different blocks, with defined terminal structures and with narrow polydispersities [9].

However, there are two main drawbacks that limit the use of NCA on an industrial scale. Firstly, with phosgene or its derivatives, very poisonous educts are needed for the synthesis of NCA. Secondly, the storage of NCA is very difficult, due to its sensitivity to moisture and heat. Thus, alternative monomers for the large-scale synthesis of polypeptides have been extensively studied over the years [12,13].

### 2.2. Schemes of Ring-Opening Polymerization (ROP) of α-Amino Acid N-Thiocarboxyanhydrides (NTAs)

α-Amino acid N-thiocarboxy anhydrides (NTAs), as thio-analogues of NCAs, are promising alternative monomers, due to the fact that they are tolerant to moisture and heat, and because their synthesis does not require the use of phosgene derivatives. In 1950, Aubert et al. first reported the synthesis of the NTA of glycine. As shown in Figure 4, this reaction involves two steps [23]. In the first step, the potassium salt of glycine is reacted with ethyl alkoxydithioformate to give N-alkoxythiocarbonyl glycine. Then, the NTA of glycine is obtained via the cyclization in presence of PBr_3_ or PCl_3_. This method is most widely used to synthesize NTA monomers, and many kinds of amino acid NTA were prepared over the next few decades using this method [11]. 

NTAs were usually used for the stepwise syntheses of polypeptides, but rarely for polymerizations before 2000. The ROP of NTAs was first investigated by Kricheldorf et al. in the 1970s. They initiated the polymerizations of several amino acid NTAs with a primary amine, and demonstrated that the polymerization underwent the normal amine mechanism (NAM) as in primary amines-initiated NCA polymerization, as shown in Figure 5 [11]. However, they thought that NTAs were not promising candidates for the preparation of high molar mass polypeptides, because of the lower yields and lower DPs [24]. Jun Ling’s groups believe that a low reaction temperature and inappropriate solvents are also responsible for the uncontrollable polymerization. Therefore, they developed NTA polymerization in polar solvents in a controlled manner, using Tyr-NTA and DOPA-NTA as examples. This polymerization produced polypeptides at a high yield (over 80%) and with predictable molar masses. However, the DPs of the products were below 50 [25]. There is still a great challenge to improve the controlled NTA polymerization in polar solvents. Carbonyl sulfide, a toxic gaseous compound, is released during the ROP of NTAs, which also limits the utilization of NTAs at a larger scale.

The ROP of N-substituted NTA (NNTA), which produces polypeptoids, was quite well-controlled, and many kinds of polypeptoids were synthesized with high yields and predictable MW because of the absence of N-H and the better solubility of the polypeptoids [11].

### 2.3. Polymerization of N-Phenoxycarbonyl Amino Acids (NPCs)

NPCs are another class of monomers that can be used for polypeptide synthesis [13]. Inspired by Kricheldorf’s synthesis of polyamides using the monomer R-(N-aryloxycarbonyl)amino-ω-carboxylalkane, in 2008, Endo et al. first synthesized several amino acid NPCs and studied their polymerization [26,27]. NPCs are also called amino acid urethane derivatives (UDs) if hydrogens at the phenoxy group of the NPC are substituted. Two routes are mainly used to prepare NPCs. One is via a reaction between amino acids and phenyl chloroformate. The other one is via a reaction between amino acids and diphenyl carbonate (DPC) in the presence of an excess base (Et_3_N or tetrabutylammonium hydroxide) (Figure 6). Using these two methods, Endo’s group prepared a large variety of NPCs over the course of 10 years, and proved that NPCs are promising alternatives for polypeptide synthesis [13].

The controlled polymerization of these NPCs can be achieved via a reaction in DMAc solution at 60 °C with a primary amine as an initiator, which can form polypeptides with predictable MW and narrow MW distribution (less than 1.2). The described polymerization may follow the same mechanism as the polymerization of NCAs, because they observe the in situ formation of NCAs using ^1^H NMR before and during the polymerizations (Figure 7) [13].

In the following, the history of Endo’s research on the polymerization of NPCs will be briefly introduced, because the synthesis allows for the production of polypeptides efficiently without using hazardous chemicals such as phosgene. In 2008, Endo et al. investigated the potential of three activated urethane-type derivatives of λ-benzyl-L-glutamate as monomers for polypeptide synthesis [26]. They studied the influence of different factors on the polymerization reaction, including solvents, temperature, monomer concentration, and monomer types. They concluded that polar solvents such as DMAc and elevated temperatures (60 °C) were essential for polymerization. Among the three urethanes, (N-phenyloxycarbonyl-λ-benzyl-L-glutamate, 4-chlorophenyloxycarbonyl-λ-benzyl-L-glutamate, and 4-nitrophenyloxycarbonyl-λ-benzyl-L-glutamate), 4-nitrophenoxycarbonyl-λ-benzyl-L-glutamate was the most reactive urethane for producing poly(BLG) efficiently, as was expected due to the high electron deficiency of the nitrophenoxycarbonyl group. At that time, they did not add any initiators, and terminal structures were also not defined. Later they used N-(4-nitrophenoxycarbonyl)-γ-benzyl-L-glutamate as a monomer, and butylamine as an initiator for the polymerizations, and they revealed that butylamine was incorporated into the terminal end of poly(BLG), while the other terminal end was endowed with an amino group [27]. Later, in 2013 and 2014, they reported the synthesis of a series of hydrophilic polypeptides, including poly-L-leucine, poly-L-phenylalanine, poly-L-valine, etc., as well as a series of hydrophobic polypeptides, including poly-L-serine, poly-L-cysteine, poly-L-asparagine, etc., with predictable molecular weights, narrow molecular weight distributions, and well-defined terminal structures [28,29]. Based on these years of research, Endo and Sudo have recently written a very good review, in which more knowledge about the history and development of NPCs can be found [13]. The pros and cons of the three different monomers are shown in Table 1.

## 3. Polypeptides for Anticancer Drug Delivery

Cancer is the second leading cause of death worldwide. Current drug-loaded nanocarriers used for cancer treatment can only mitigate the adverse effects, but they cannot improve the therapeutic efficacy of anticancer drugs. The main reason for this low therapeutic efficacy is that it is difficult to transfer the anticancer drugs into the target tumor cells as free molecules [30]. In 2017, Shen et al. analyzed the typical cancer-drug-delivery process of intravenously administered drug-loaded nanocarriers, and they concluded that the delivery involves five steps: blood circulation, accumulation at the tumor site, tumor internal penetration, cellular internalization, and intracellular drug release; this was named the CAPIR cascade (Figure 8) [31]. They believe that a high efficiency at every step is crucial to ensure a high overall therapeutic efficiency.

To achieve a high efficiency in the corresponding steps, a nanocarrier must have three specific capabilities: good drug-carrying characteristics, suitable surface properties, and sufficient diffusivity. For drug carrying, it should first hold the drug tightly before entering cancer cells, but it must release the drug quickly once inside the cells. For the nanocarriers’ surface, it should also be stealthy for a long time during blood circulation, to give time for tumor accumulation; on the other hand, it must be able to interact with tumor cells for efficient cellular uptake after reaching the tumor cells. For diffusivity, the nanocarrier must have sufficient mobility to penetrate deep into the tumor so that it can reach tumor cells remote from the blood vessels [30]. To meet these requirements, nanocarriers should exhibit different or even opposite properties at different CAPIR steps. However, they can be easily grouped into three transitions of the nanoproperties: stability, surface, and size, also known as the 3S transitions [31].

Polypeptides can be conveniently endowed with stimuli responsiveness by introducing natural amino acid residues with innate stimuli-responsive characteristics, or by introducing responsive moieties to their side chains using simple conjugating methods, which makes them suitable carriers of anticancer drugs to fulfill the 3S transitions [2]. In the following, we will review the polypeptide carriers for anticancer drug delivery which fulfill the 3S transitions.

### 3.1. Stability Transition Polypeptide Systems

The stability transition means that nanocarriers must remain stable enough during blood circulation, but that they should release the drug quickly after entering the tumor cells. To achieve the transition from stable to unstable, many stimuli-responsive polypeptide-based nanocarriers have been prepared over the past 10 years. There are two main strategies to construct stimuli-responsive polypeptide-based nanocarriers. The first is to introduce blood-stable but intracellular-labile bonds between polypeptide–drug conjugates, including acid-labile bonds (such as hydrazine [32,33] or benzoic imine [34]) responding to lysosomal acidity, and glutathione (GSH)-sensitive disulfide bonds responding to elevated GSH levels in tumor cells [35,36,37]. Our group synthesized a tetra-doxorubicin-tailed polyethylene glycol via benzoic-imine bond linkage, which can self-assemble into a pH-sensitive prodrug micelle. The micelle can quickly release doxorubicin in tumor sites to exert anticancer activity according to the experiments (Figure 9) [34]. Li et al. conjugated disulfide-containing camptothecin to poly(L-glutamic acid)-graft-methoxy poly(ethylene glycol) to create an amphiphilic biodegradable prodrug, which could self-assemble into micellar nanoparticles and encapsulate doxorubicin. The drug-release experiment proved that camptothecin could be released quickly in a GSH water solution at a cytosolic concentration (10 mM), which is a promising GSH-triggered drug-release system [36].

The other strategy is to prepare polypeptide drug carriers that can disintegrate under some special stimulation conditions [2,38], including a low pH value [39,40,41], low oxygen concentration [42,43], high GSH concentration [44], or a high concentration of reactive oxygen species (ROS) [45,46]. Zhong et al. synthesized a pH-responsive block polymer consisting of PEG and poly(asparagyl diisopropylethylenediamine-co-phenylalanine), which can self-assemble into nanovesicles to encapsulate hydrophilic tirapazamine and the hydrophobic photosensitizer dihydrogen porphin [47]. These nanovesicles would disassemble when incubated in a solution of 10 mM phosphate-buffered saline (PBS) at pH 5.0 for 24 h, due to the pH sensitivity of the diisopropylethylenediamine segment, which can enhance the release of the encapsulated active ingredient. Hoang et al. synthesized a ROS-responsive poly(ethylene glycol)-poly(methionine) and prepared micelles via self-assembly with a hydrophobic pro-oxidant drug, piperlongumine [45]. The increased ROS content in cancer cells triggered a hydrophobic to hydrophilic transition of the polypeptide, which led to the disassembly of the micelles, and consequently, to efficient drug release, increasing the anticancer efficiency.

### 3.2. Surface Transition Polypeptide Systems

The surface transition means that the carriers should remain stealthy during blood circulation, but after they reach the tumor tissue, they should be sticky toward tumor cells for efficient internalization. To achieve a stealthy-to-sticky transition, the surface properties of polypeptide carriers could undergo several changes such as PEGylation/dePEGylation, and changes in the surface charge. Both alternatives are intensively studied in recent years.

Poly(ethylene glycol) (PEG) has been demonstrated to give nanocarriers a stealth property. Nanocarriers with a PEG shell will show long blood circulation times, which is essential for the nanocarriers’ passive tumor targeting/accumulation, through their enhanced permeability and retention (EPR) effects [48]. However, this stealth layer also slows the nanocarriers’ cellular uptake, which limits their therapeutic effect. Therefore, many polypeptide carriers with detachable PEG shells have been designed in recent years [49,50]. Jiang et al. prepared two kinds of cisplatin-loaded poly(glutamate-lysine) complex nanoformulations with detachable PEG grafted onto lysine segments through two different bridged chemical bonds, which are responsive to specific tumor tissue microenvironments, including low pH and matrix metalloproteinases-2/9 [49]. The nanoformulations with PEG showed a prolonged circulation time in the blood and increased drug accumulation in the tumor tissue compared to the nanoformulations without the PEG segment. After arriving at the tumor tissues, the nanoformulations also showed enhanced cell uptake and cytotoxicity, due to the cleavage of the bridged chemical bond between PEG and polylysine (Figure 10). Wu et al. fabricated a multivalent amphiphilic peptide dendrimer to form nanoparticles with two siRNAs and linked the aldehyde-alkylated aptamer–PEG to the amino group of the nanoparticles through an acid-sensitive Schiff base, to construct tumor-activatable nanoparticles [50]. After arriving at the tumor tissues, the PEG layer of the nanoparticles was removed due to the low pH values, which facilitated siRNA penetration into the tumor tissues.

Studies have shown that nanocarriers that are slightly negatively charged or uncharged can remain stable in the circulatory system, while positively charged nanoparticles are more likely to promote cellular internalization and transcytosis for deep tumor penetration [51]. Therefore, charge-reversal nanocarriers based on polypeptides have been intensively studied over the years [52,53,54,55,56,57,58,59]. Li et al. fabricated a surface charge-reversible nonviral gene vector with PEG, polypeptide, and PEI [60]. Owing to the protonation of PEI, the vector was negatively charged during circulation, but positively charged once it entered the tumor tissue, which facilitates both long circulations in the bloodstream as well as easy cell uptake. Qu et al. synthesized pH-sensitive polypeptide hybrid terpolymers, poly (lysine-co-N,N-bis(acryloyl) cystamine-co-dimethylmaleic anhydride), which can self-assemble into spherical nano-micelles with a negative surface charge under normal physiological conditions [61]. After arriving at the slightly acidic tumor tissues, the surface charge of the micelles switched from negative to positive, due to the protonation of lysine residues to enhance cellular uptake. Later, the same group also prepared dual pH and redox-responsive cross-linked polypeptides based on poly(L-lysine-co-N,N-bis(acryloyl)cystamine-co-γ-glutamic acid), which can self-assemble into nanoparticles with negatively charged surfaces under physiological conditions [62]. The surface charge of the nanoparticles can switch to positive in a slightly acidic tumor extracellular environment because of the protonation of lysine residues, which increases the cellular uptake efficacy.

### 3.3. Size-Transition Polypeptide Systems

Size transition requires a carrier size of around 100 nm during blood circulation, but this size should be reduced to approximately 30 nm when arriving at the tumor tissue. Studies have shown that nanoparticles with a size smaller than 30 nm or larger than 200 nm are easily cleared, while nanoparticles around 100 nm are most likely to achieve long circulation times, which are crucial for the passive targeting of EPR effects in tumor tissues. However, this size is too big for the nanocarriers to diffuse into the tumor, and only nanoparticles with a size that is smaller than 30 nm exhibit a high penetration ability [63]. Therefore, polypeptide carriers that can realize a size transition have been intensively studied over the years [64,65,66,67,68,69]. Cun et al. prepared a size-switchable nanoplatform by conjugating small dendrigraft polylysine (DGL) to poly(ethylene glycol)-poly(caprolactone) micelles via a matrix metalloproteinase 2-sensitive peptide [66]. The nanoplatform had an initial size of 100 nm and a nearly neutral charge, which was suitable for a long circulation time. After arriving at the tumor tissues, small DGL/DOX nanoparticles, which were approximately 30 nm in size, were rapidly released from the nanoplatform due to the cleavage of enzyme-sensitive peptides, thus enhancing the penetration of tumor cells. 

The size transition is normally accompanied by other transitions such as dePEGylation [70]. For example, Chen et al. prepared shell-stacked nanoparticles for deep penetration into solid tumors based on polypeptides [71]. They used positively charged polylysine-polycysteine-polyphenylalanine as core materials to support DOX, and negatively charged PEG-poly(lysine-dimethylmaleic anhydride) as shell materials. The nanoparticles had a size of 145 nm and a zeta potential of −7.4 mV, which led to a long circulation time. When arriving at the acidic tumor tissue, their size was reduced to 40 nm and their surface charge was reversed up to 8.2 mV, which enhanced tumor penetration and uptake by cells in deep tumor tissue. After entering the tumor cells, the cleavage of the disulfide bond of the carrier reduced the stability of the carrier, thereby accelerating the release of the drug, as shown in Figure 11. Therefore, this polypeptide-based carrier also seems to fulfill all of the 3S transitions. Some examples of different promising transition polypeptide systems are listed in Table 2, categorized into the most commonly used categories.

### 3.4. Polypeptide Systems Targeting Tumor Tissues

For the anti-tumor active agents that could have direct anti-tumor effects on tumor tissues, the steps of cell internalization and release are not required, which significantly simplifies the design idea for the polypeptide carrier. For photothermal therapy, the delivery of the photosensitizer, which can generate heat for the thermal ablation of cancer cells upon NIR laser irradiation, into tumor tissues, is sufficient for treatment, and thus only the circulation and accumulation steps are needed [31,73,74]. Therefore, many stable polypeptide carriers for photosensitizer delivery have been extensively studied in recent years [75,76]. Huang et al. fabricated photothermal therapeutic nanocarriers by conjugating indocyanine green dye with self-assembled polypeptides via chemical bonding. The drug-loaded nanocarrier showed a diameter of around 50 nm, and had an improved degree of tumor accumulation and photothermal effect compared to the free dye [75]. 

Photodynamic therapy (PDT) is a similar treatment in which photosensitizers are used to generate reactive oxygen species to induce oxidation stress to kill cancer cells; however, these photosensitizers still need to be delivered to the cancer cells. However, the therapeutic effect of PDT is frequently limited due to the hypoxic nature that is characteristic of many solid tumors [77]. Thus, numerous efforts have been made to increase the oxygen content in tumors to enhance PDT efficacy [72]. Our group is now focusing on the synthesis of perfluorodecalin-filled oxygen carriers, using polypeptides as encapsulating shell materials, which allow for fast gas exchange [78]. The capsule is expected to be used to improve the hypoxic condition of tumor tissue to enhance photodynamic therapy in the future. 

## 4. Critical Evaluation and Future Perspectives

Even though polypeptide drug delivery systems have already achieved good results in preclinical studies, there is still a significant lack of clinical trials using these systems. Obviously, there is a large gap between preclinical studies and their translation to the clinic. The lack of consistency in reporting preclinical studies using drug delivery systems has prevented a systematic assessment of these studies [79]. For the effective application of nanotechnology in the clinical environment, researchers should be able to compare new data with previously published results in a reliable and meaningful way. This requires a standard of specific information that needs to be reported, including material characterization, biological characterization, and so on, so that quantitative comparisons, meta-analyses and in silico modeling can be conducted and facilitated [80]. In addition, the lack of studies on the interaction between delivery systems and blood hampers clinical translation. When entering the blood, drug delivery systems are shielded with a biomolecular corona, which changes the physicochemical properties of the delivery systems (e.g., the shape, size, charge, and surface coatings), and even causes mistargeting [81,82]. Another crucial issue is the low level of accumulation of the delivery systems in tumors, and our lack of understanding of its mechanisms. According to an analysis by Wilhelm et al., only about 0.7% of the injected dose is actually accumulated in the tumor, which is extremely low relative to the efficiency index of most therapeutic platforms [83]. The tumor accumulation of nanoparticles via the EPR effect seems to be questionable [84,85]. More and more evidence suggests that the dominant mechanism of the extravasation of nanoparticles into solid tumors may be through an active process of endothelial transcytosis, rather than the currently established mechanism of passive transport via inter-endothelial gaps. Scrutinizing these pathways will unlock new approaches toward improving tumor accumulation, and this should change the way in which we think about delivering drugs to tumors using biomaterials [79].

In summary, this review introduced the synthesis of polypeptide through three kinds of activated amino acid monomers, including NCAs, NTAs, and NPCs, as well as their biomedical applications as anti-tumor drug carriers for tumor treatment in detail. Despite the limited clinical products of nanocarriers based on polypeptides to date, we believe that with advances in peptide synthesis, the introduction of new design concepts and analytical methods, and a better understanding of tumors, the efficiency and efficacy of nanocarriers based on polypeptides will be further improved, thereby facilitating their clinical translation.

## Figures and Tables

**Figure 1 ijms-23-05042-f001:**
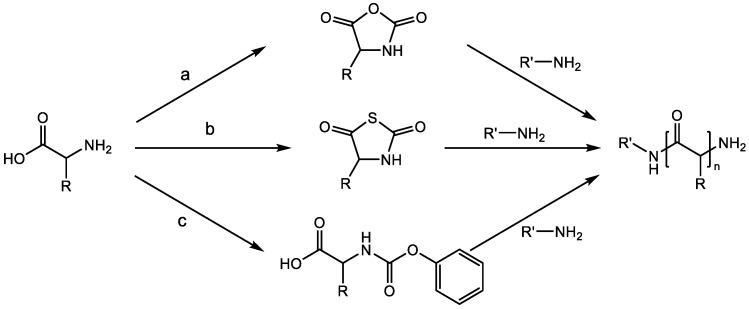
Polymerization of three kinds of activated monomers. (a) Phosgene or its derivatives; (b) Ethyl alkoxydithioformate and PCl_3_; (c) Phenyl chloroformate or dibenzyl carbonate.

**Figure 2 ijms-23-05042-f002:**
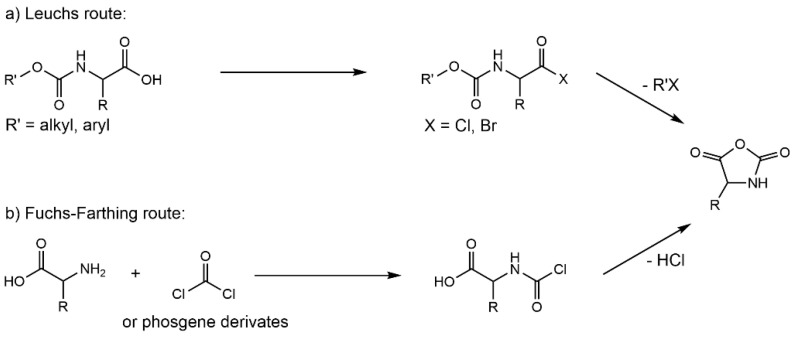
(**a**) Leuchs method to synthesize NCAs, (**b**) Fuchs–Farthing method to synthesize NCAs.

**Figure 3 ijms-23-05042-f003:**
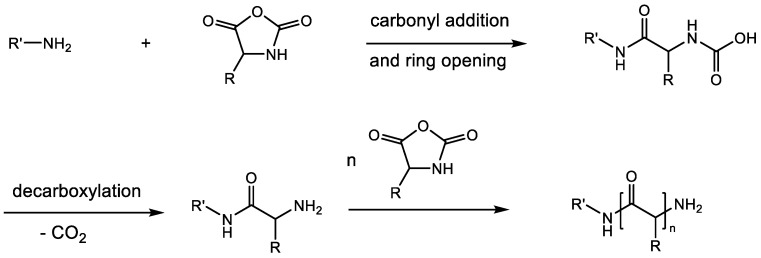
Mechanism of primary amines-initiated ROP of NCAs.

**Figure 4 ijms-23-05042-f004:**
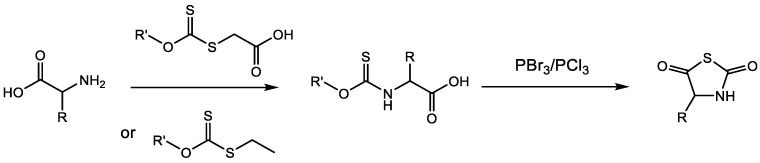
A common route to synthesize NTAs.

**Figure 5 ijms-23-05042-f005:**
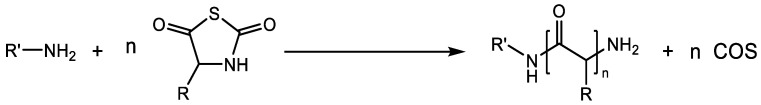
Synthesis of polypeptides via ring-opening polymerization of NTA.

**Figure 6 ijms-23-05042-f006:**
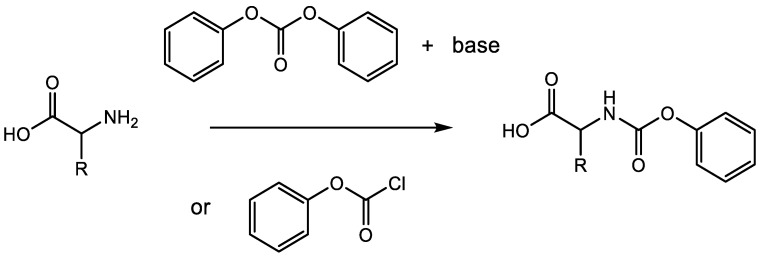
Synthetic methods to prepare NPC monomers.

**Figure 7 ijms-23-05042-f007:**
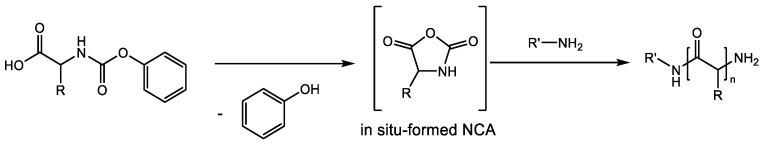
Synthesis of polypeptides via polymerization of NPCs.

**Figure 8 ijms-23-05042-f008:**
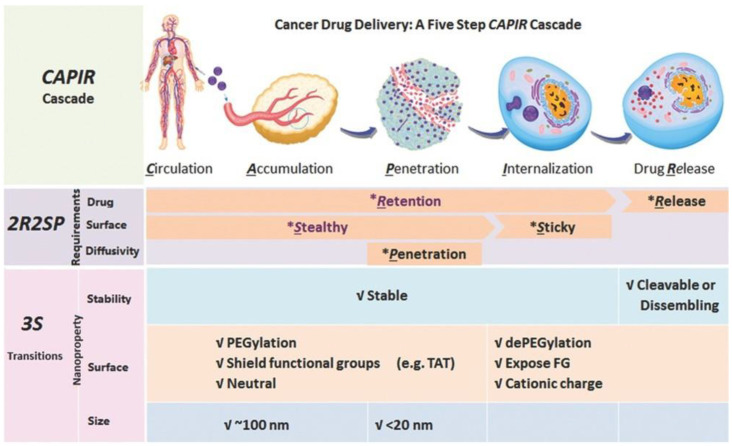
Summary of the 2R2SP requirements and the 3S transitions in the CAPIR cascade for a nanocarrier to have high overall drug-delivery efficiency. Reprinted with permission from Ref. [31]. Copyright © 2017 WILEY-VCH Verlag GmbH & Co. KGaA, Weinheim.

**Figure 9 ijms-23-05042-f009:**
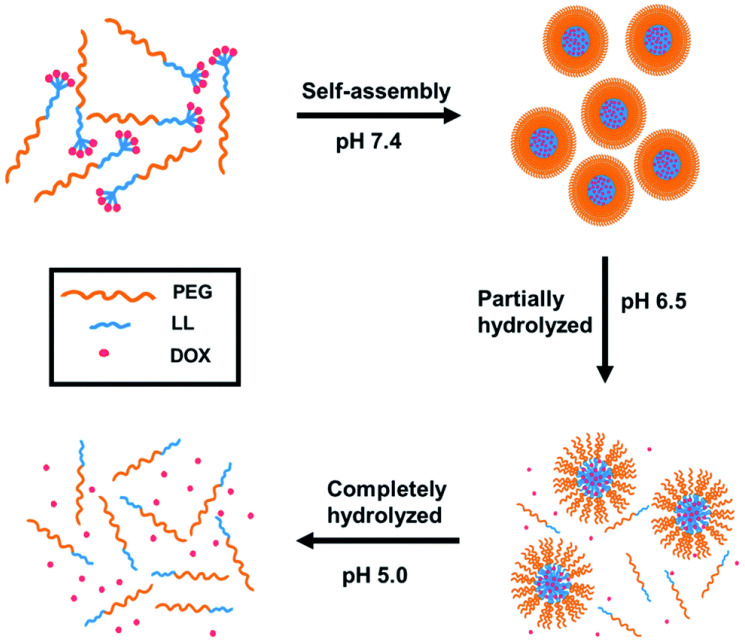
Schematic illustration of self-assembly and the low pH-triggered DOX release from the self-assembled prodrug micelle. Reprinted with permission from Ref. [34]. Copyright © 2016 The Royal Society of Chemistry.

**Figure 10 ijms-23-05042-f010:**
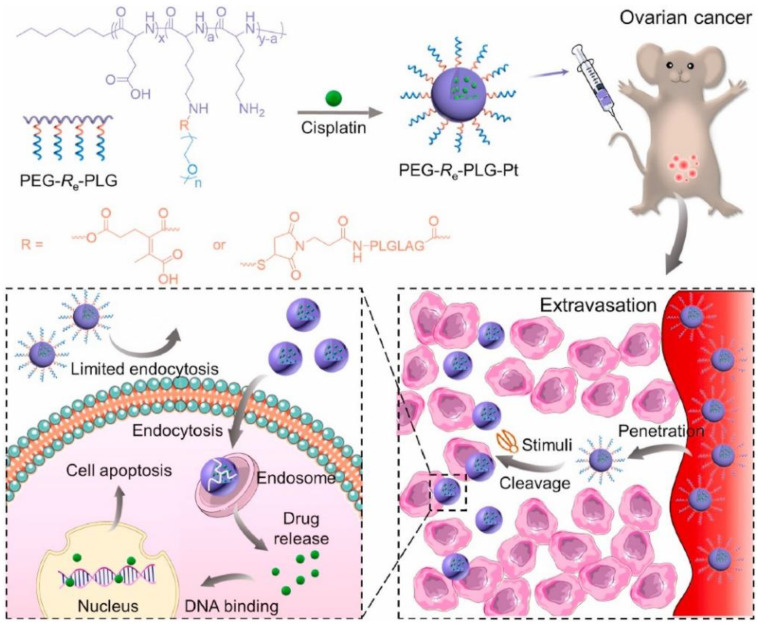
Schematic illustration for preparation of PLG-CDDP nanoformulations with detachable PEG response to tumor microenvironments for enhanced therapy of peritoneal metastasis of ovarian cancer. Upon reaching the tumor tissue, dePEGylation is triggered by acidic pH or overexpression of MMP, and the nanoformulations without a PEG shell enter and deliver the drug to the cancer cells more effectively, leading to improved antitumor efficacy. Reprinted with permission from Ref. [49]. Copyright © 2021 The Authors.

**Figure 11 ijms-23-05042-f011:**
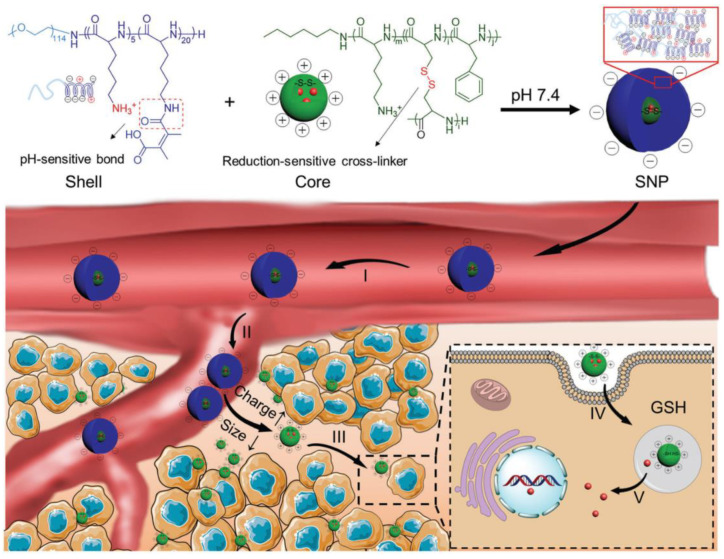
Formation and deep tumor penetration of shell-stacked nanoparticle (SNP). SNP is composed of a PEGylated and negatively charged shell, and a positively charged and disulfide-cross-linked polypeptide core, through electrostatic interaction. Owing to the tumor-microenvironment-mediated multi-transformations, SNP could lead to a perfect cascade of drug delivery. (I) Long circulation. (II) Enhanced accumulation. (III) Deep penetration. (IV) Promoted internalization. (V) Accelerated drug release. Reprinted with permission from Ref. [71]. © 2017 WILEY-VCH Verlag GmbH & Co. KGaA, Weinheim.

**Table 1 ijms-23-05042-t001:** Pros and cons of different monomers.

Monomers	Advantages	Disadvantages
NCAs	High reactivity Easy synthesis	Phosgene derivatives used to synthesize NCAs are lethal or toxic. Unstable in moisture
NTAs	More stable under moisture and heat than NCAs. Easy isolation and storage.	Educts used to prepare NTAs are expensive. Toxic gaseous compound, carbonyl sulfide, is formed during polymerization.
NPCs	More stable under moisture and heat than NCAs. Easy synthesis, isolation, and storage.	The yield of polymerization is slightly lower than that of NCA.

**Table 2 ijms-23-05042-t002:** Examples of different transition polypeptide systems for anti-tumor drug delivery.

Polypeptides	Drugs	Stimuli-Responsiveness	Transition Types	Ref.
polylysine	doxorubicin	pH	stability transition	[34]
polylysine	gemcitabine	GSH	stability transition	[35]
polyglutamic acid	camptothecin	GSH	stability transition	[36]
decapeptide consisting of leucine and lysine	doxorubicin	pH	stability transition	[40]
polymethionine	piperlongumine	ROS	stability transition	[45]
polyglutamate	doxorubicin	hypoxia	stability transition	[42]
polylysine dendrigraft	curcumin and doxorubicin	pH and enzyme	stability transition	[41]
polyaspartate and polyphenylalanine	tirapazamine	light and hypoxia	stability transition	[47]
polyglutamic acid	cisplatin	pH and enzyme	surface transition	[49]
polylysine and polycysteine	nitric oxide donor and doxorubicin	pH and light	surface transition	[56]
polyaspartate	siRNA	pH	surface transition	[60]
polylysine and polyleucine	doxorubicin	pH	surface transition	[52]
polylysine	doxorubicin	pH	surface transition	[58]
polyaspartate	paclitaxel and curcumin	pH	surface and size transition	[54]
polylysine	doxorubicin	pH	stability and surface transition	[53]
polylysine and polycysteine	doxorubicin	pH and GSH	stability and surface transition	[57]
polylysine	doxorubicin	pH and GSH	stability and surface transition	[61]
polylysine	triptolide and doxorubicin	pH and GSH	stability and surface transition	[59]
polylysine and polyglutamic acid	doxorubicin	pH and GSH	stability and surface transition	[62]
polyglutamic acid	doxorubicin	pH	size transition	[65]
polylysine	doxorubicin	enzyme	surface and size transition	[66]
hexadecapeptide consisting of lysine and glutamic acid	SN-38	pH	size transition	[69]
polylysine and polycysteine	doxorubicin	pH and GSH	3S transition	[71]
polylysine and polyglutamic acid	cisplatin	pH and GSH	3S transition	[70]
polyaspartate	photosensitizers and hemoglobin	no stimuli	no transition	[72]

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
