# Peer review of "Syntheses of Polypeptides and Their Biomedical Application for Anti-Tumor Drug Delivery"

_ijms, 2022, doi:10.3390/ijms23095042_

Round 1

Reviewer 1 Report

My suggestions to the authors are listed below as “General comments” and “Specific Comments”.  

General comments:

- The present review work can only be accepted in the form of a “mini-review” paper. The term could be included in the abstract or in the title.

- English language and style need revision. Some parts of the text are not clearly understood and there is a misuse of the present and past tense, and active and passive voice throughout the text; syntax errors can be read too. Thus, it is very difficult to follow the text flow and really understand what the authors try to convey.

- Text flow must also be improved. Some paragraphs are poorly organized and make difficult to follow the authors’ ratiocination.

- In some parts appropriate references to related works are missing. See Specific comments.

- The opinion of the authors is missing. Review articles should state the current overview of the literature to a specific area, but also to provide the opinion of the expert on that topic.

- Introduction section needs to be re-written. Only some info about the synthetic pathways of the different types is presented. This info can be transferred to section 2. Please revise Introduction section and give some info about polypeptides as drug delivery platforms in general and also give some info about their use in the treatment of tumors. Maybe highlight their superiority over other drug delivery platforms. Also, in this section; some info about the way the present review is conducted should be added. Which literature sources were used? What is the year gap that the present review covers? What search terms were used? How many papers did you find? Were there eliminations? Perhaps you should consider including a study selection flow diagram (the time period of publication and the number of documents need to be marked) and the study characteristics table (listing specific document data). A column diagram showing the number of studies by year can be also included.

- Section 2 is gives rather historical info about the sunthetic routes, than essential info about them. Maybe considering adding a table comparing the different pathways could elucidate the differences. Also, a comparison between NCAs, NPCs and NTAs should also be performed. Perhaps a table concentrating the properties, pros and cons of NCAs, NPCs and NTAs could help here.

- Section 3 is well written and gives important info. However, it needs improvement in terms of paragraph organization and text flow. In my opinion, the most valuable info you give is Table 1. Thus, I believe you should enrich more the table with additional studies and info.

- Please include an “Author Contribution” section.

Specific comments:

- line 2, Title: Please correct typo

- lines 28 – 33: Please rearrange references to appear after each corresponding example

- lines 37 – 38: Please add appropriate reference for the statements here

- lines 44 – 46: Please add appropriate reference for the statements here

- lines 64 – 65: Please add appropriate reference for the statements here

- line 73, Figure 2: Please improve figure quality

- lines 80 – 83: Please add appropriate reference for the statements here

- lines 89 – 91: Please add appropriate reference for the statements here

- lines 112 – 116: Please add appropriate reference for the statements here

- line 176: Why do you focus on Endo’s research? Explain in text.

- lines 201 – 203: Please add appropriate reference for the statements here

- lines 209 – 215: Please add appropriate reference for the statements here

- line 223, Figure 8: Please improve figure quality

- line 274: Please add appropriate reference for the statements here

- lines 278 – 279: Please add appropriate reference for the statements here

- lines 302 – 305: Please add appropriate reference for the statements here

- lines 350 – 351: On what basis was the selection of these studies included in Table 1 performed?

Author Response

Dear Reviewer,

We want to thank you for the thoughtful comments and recommendations and we respond to all issues one by one. The comments by reviewer 1 and 2 are listed, each followed by our response within the attached document.

All changes in the manuscript are indicated by yellow marking. We hope that, with the additions and changes made, our manuscript is now acceptable for publication in International Journal of Molecular Sciences.

We are looking forward to further comments and recommendations.

With kind regards                                                                                           

Huayang Feng

Ph.D. candidate under supervision of Prof. Dr. Christian Mayer

Inst. for Physical Chemistry

University of Duisburg-Essen

D 45141 Essen, Germany

Reviewer 2 Report

General comment

The manuscript entitled “Synthesis of polypeptide and its biomedical application for anti-tumor drug delivery” aims to summarize the recent evidence on challenges and perspective of polypeptide synthesis in biomedical application, in particular, in anti-tumor drug delivery. The manuscript is well written, clear and fluent, being an interesting and summarizing work. Overall, few minor corrections are required.

  • Minor English corrections and typos should be revised and corrected along the text.
  • The section between 92-108 should be revised as it seems less clear than other section in the text.
  • Figure 8-9-10 have to be uploaded with increased resolution
  • “for example” is repeated often in the manuscript, being redundant. Please revise.

Author Response

(The authors gave the same response as above.)

Round 2

Reviewer 1 Report

Thank you for trusting my suggestions!